# Integration of Network Pharmacology, Molecular Docking, and In Vitro Nitric Oxide Inhibition Assay to Explore the Mechanism of Action of Thai Traditional Polyherbal Remedy, Mo-Ha-Rak, in the Treatment of Prolonged Fever

**DOI:** 10.3390/ph18101541

**Published:** 2025-10-13

**Authors:** Chinnaphat Chaloemram, Ruchilak Rattarom, Anake Kijjoa, Somsak Nualkaew

**Affiliations:** 1Doctor of Philosophy in Pharmacy Program, Faculty of Pharmacy, Mahasarakham University, Kantharawichai, Maha Sarakham 44150, Thailand; chinnaphat.med@gmail.com; 2Pharmaceutical Chemistry and Natural Product Research Unit, Faculty of Pharmacy, Mahasarakham University, Kantharawichai, Maha Sarakham 44150, Thailand; rujiluk.r@msu.ac.th; 3School of Medicine and Biomedical Sciences (ICBAS) and CIIMAR, Universidade do Porto, Rua de Jorge Viterbo Ferreira 228, 4050-313 Porto, Portugal; ankijjoa@icbas.up.pt

**Keywords:** prolonged fever, Mo-Ha-Rak, network pharmacology, molecular docking, anti-inflammatory, NO inhibitory activity

## Abstract

**Background:** Prolonged fever (PF) is one of the most challenging clinical conditions due to its complex molecular mechanisms and limited effective treatments. **Objective:** The current study aimed to explore the mechanism of action of Mo-Ha-Rak (MHR), a Thai traditional polyherbal remedy, in PF treatment. **Methods:** Integration of network pharmacology, molecular docking, and inhibition of nitric oxide (NO) production in LPS-induced RAW264.7 macrophages approaches were used. **Results:** The study identified 86 potential active compounds, 131 potential therapeutic targets, and 9 hub genes for MHR. Key targets with the highest degree of connectivity in PF, including TNF, IL6, IL1B, PTGS2, STAT3, and NFKB1, are closely associated with arachidonic acid metabolism pathways, which play critical roles in infections, inflammation, cell proliferation, and apoptosis in the PF microenvironment. Molecular docking analysis suggested that core compounds exhibited strong binding affinities for four key targets, viz. TNF, IL6, IL1B, and PTGS2, with binding energies ranging from −4.1 to −9.8 kJ/mol. MHR exhibited dose-dependent reduction of NO production at concentrations of 10–100 µg/mL. Among the biomarkers of MHR tested, ellagic acid, loureirin A, resveratrol, and rhein showed potential to inhibit NO production. **Conclusions:** This study demonstrates that MHR exerts its therapeutic effects on PF through a complex network of multiple compounds, targets, and pathways. These findings highlight the mechanisms of PF and the role of MHR in modulating the arachidonic acid metabolism pathway, which underlies the development of fever.

## 1. Introduction

Prolonged fever (PF), or a fever of unknown origin, which can occur in both children and adults, is one of the most challenging clinical conditions [1,2]. PF was first described in 1961 as a prolonged febrile illness, characterized by a temperature of 38 °C or higher and lasting for three weeks or longer, without an established cause despite a one-week inpatient evaluation [3]. The common causes of PF include infections (20–40%), malignancies (20–30%), noninfectious inflammatory diseases (10–30%), miscellaneous conditions (10–20%), and undiagnosed cases [4,5,6,7,8,9,10]. Additionally, drug-induced fever accounts for 1–3% of PF cases [4]. Given these complexities, there is a pressing need for the development of safe and effective treatments to reduce reliance on chemotherapy in PF management.

Although common fevers that require no treatment are of little clinical concern, PF involves significant implications for patient care. As a type of fever, PF tends to be annoying and challenging, which may require repeated care and extensive treatments. Moreover, a long-lasting fever may aggravate immune function and can be maladaptive, possibly by inhibiting the apoptosis of immune cells and perpetuating a pro-inflammatory cytokine response [11]. Additionally, elevated temperatures also exacerbate brain tissue damage [12]. Furthermore, PF may independently lead to poor neurological outcomes and increased mortality [11,12]. Therefore, PF should be one of the topics for the top research agenda, with timely evaluation and treatment warranted.

Polyherbal remedies have long been used in Thai traditional medicine for the treatment of a myriad of diseases. Traditional formulations often consist of various herbs, and each serves distinct therapeutic functions. Mo-Ha-Rak (MHR) is a polyherbal remedy composed of 21 medicinal plants, and is used by Thai traditional practitioners for the treatment of PF, poisonous fever (high fever or fever with rash), accompanied by drowsiness, internal body heat, restlessness, delirium, unconsciousness, and internal body inflammation. MHR remedy is prepared by cutting each of the open-air dried 21 plants (with the ratio indicated in Table 1) into small pieces (1–2 cm). The plant materials are thoroughly mixed, transferred into a clay recipient, and then the water (1:5 w/v) is added. After boiling for 30 min, the mixture is allowed to cool down to room temperature and then filtered with a cheesecloth. The filtrate is an MHR remedy. The dosage for adults is one glass in the morning before a meal. The dosage for children is proportionately adjusted according to their age [13]. Several plants in the MHR remedy have been reported to possess antipyretic and anti-inflammatory properties, as summarized in Table 1. Although MHR has been used to treat patients in Thai traditional medicine for a long time, there has been no scientific investigation or clear evidence supporting its mechanism of action because of its complex chemical composition.

Network pharmacology is a systems-level approach that integrates bioinformatics, cheminformatics, and systems biology to dissect the complex interactions between bioactive compounds, molecular targets, and disease-related pathways. In the context of traditional herbal formulas, network pharmacology enables the prediction of putative targets for multiple compounds, construction of compound–target and target–disease interaction networks, and identification of hub nodes (key proteins) via topological analysis (e.g., degree, betweenness, closeness). Enrichment analyses (i.e., Gene Ontology and KEGG pathways) are then applied to infer the biological processes and pathways potentially modulated by the formula. This approach facilitates hypotheses on how herbal mixtures may act via multiple components and multiple targets in a coordinated manner. Recently, network pharmacology has been successfully applied to elucidate the mechanisms of multi-component herbal formulas in many traditional medicines [14,15,16,17,18].

As mentioned above, MHR is a polyherbal formulation whose pharmacological properties and active constituents have yet to be fully characterized. Thus, this study aims to elucidate the potential mechanisms of action of MHR in the treatment of PF, with a particular focus on its anti-inflammatory properties. Consequently, the research focused on the identification of the active components, core therapeutic targets, and potential synergistic mechanisms involved. Since the onset of fever is primarily triggered by cytokines involved in the inflammatory process, understanding these mechanisms can facilitate the evaluation of relevant pharmacological effects and guide the selection of appropriate formulations for further in vivo pharmacological studies, especially those related to anti-inflammatory activity. MHR was also assayed for its capacity to inhibit nitric oxide (NO) production to validate the aforementioned hypothesis.

**Table 1 pharmaceuticals-18-01541-t001:** The herbal constituents of MHR and their reported pharmacological activities.

Botanical Name	Part Used	Ratio (% *w*/*w*)	Pharmacological Activities
*Azadirachta indica* A.Juss.	Petiole	4	-
*Bridelia ovata* Decne.	Leaf	4	-
*Capparis micracantha* DC.	Root	4	Antipyretic [19,20], Anti-inflammatory [21]
*Cassia fistula* L.	Pulp	12	Antipyretic [22], Anti-inflammatory [23]
*Clerodendrum indicum* (L.) Kuntze	Root	4	Antipyretic [20], Anti-inflammatory [24]
*Dracaena cochinchinensis* (Lour.) S.C.Chen	Wood	4	Antipyretic [25], Anti-inflammatory [26,27]
*Ficus racemosa* L.	Root	4	Antipyretic [19,20], Anti-inflammatory [28,29]
*Gymnopetalum chinense* (Lour.) Merr.	Fruit	4	Anti-inflammatory [30]
*Harrisonia perforata* (Blanco) Merr.	Root	4	Antipyretic [19,20], Anti-inflammatory [31,32,33]
*Ligusticum sinense* Oliv.	Rhizome	1	Anti-inflammatory [34]
*Mesua ferrea* L.	Flower	4	Anti-inflammatory [35]
*Nelumbo nucifera* Gaertn.	Stamen	2	Anti-inflammatory [36]
*Phyllanthus emblica* L.	Fruit	8	Anti-inflammatory [37]
*Pinus kesiya* Royle ex Gordon	Wood	1	-
*Tarenna hoaensis* Pit.	Wood	4	-
*Terminalia bellirica* (Gaertn.) Roxb.	Fruit	8	Antipyretic [38], Anti-inflammatory [39,40]
*Terminalia chebula* Retz.	Fruit	8	Antipyretic [41], Anti-inflammatory [42]
*Terminalia* sp. “Samo Thet in Thai”	Fruit	8	-
*Tiliacora triandra* (Colebr.) Diels	Root	4	Antipyretic [19,20], Anti-inflammatory [32,43,44]
*Tinospora crispa* (L.) Hook. f. & Thomson	Stem	4	Antipyretic [45], Anti-inflammatory [46,47]
*Vetiveria zizanioides* (L.) Nash	Root	4	Antipyretic [48], Anti-inflammatory [48,49]

## 2. Results

### 2.1. Screening of Active Compounds and MHR-Related Targets

Among the 21 plants of the MHR remedy, 147 compounds were found to have anti-inflammatory activity, and these compounds were filtered to 86 compounds with high gastrointestinal (GI) absorption and desirable drug-like properties, whereas 62 compounds were excluded due to their low GI absorption (Appendix A). Subsequently, the potential targets for each of these 86 compounds were predicted, and after elimination of duplicates, the remaining targets were consolidated to yield 965 active compound-related targets. Detailed information is in Appendix A.

### 2.2. Potential Therapeutic Targets of MHR Used in PF Treatment

Using “prolonged fever” as a keyword to retrieve data from multiple databases, 8031 targets from GeneCards, 11 from the Online Mendelian Inheritance in Man (OMIM), and 97 from PharmGKB were obtained (Appendix A). After comparison of MHR-related targets with PF-related targets, 131 common targets were identified and considered potential therapeutic targets of MHR in the treatment of PF.

### 2.3. Protein–Protein Interaction (PPI) Visualization and Modular Analysis

A protein–protein interaction (PPI) network reveals relationships among 131 potential therapeutic targets (Figure 1). This network consisted of 131 nodes and 678 edges, and its key topological metrics included an average node degree of 10.59 and a local clustering coefficient of 0.539. In this network, nodes represent individual proteins, while edges depict their interactions, with higher node degrees indicating higher roles within the network. After application of filtering criteria (Figure 2a–c), nine hub genes were identified, viz. TNF (tumor necrosis factor-α), IL6 (interleukin-6), IL1B (interleukin-1β), PTGS2 (prostaglandin-endoperoxide synthase 2), STAT3 (signal transducer and activator of transcription 3), NFKB1 (nuclear factor kappa subunit 1), HDAC1 (histone deacetylase 1), PRKCA (protein kinase C alpha), and MPO (myeloperoxidase) (Table 2). These hub genes exhibited superior network topology parameters, including degree, betweenness centrality, and clustering coefficient, designating them as core targets within the PPI network (Appendix A). Notably, TNF, IL6, IL1B, PTGS2, STAT3, and NFKB1 were identified as the top six nodal targets, demonstrating strong associations with other potential therapeutic targets and playing pivotal roles in PF treatment. Cluster analysis by the Molecular Complex Detection (MCODE) plugin identified ten cluster modules (Figure 3), with module terms detailed in Table 3. Network analysis of potential therapeutic targets identified ten protein modules (MCODE), each of which is associated with distinct biological functions. For example, MCODE1 is associated with arachidonic acid metabolism, antifolate resistance, and African trypanosomiasis, while MCODE2 is connected with purine metabolism, morphine addiction, and nucleotide metabolism. MCODE3 is involved in aldosterone-regulated sodium reabsorption, inflammatory mediator regulation of transient receptor potential (TRP) channels, and African trypanosomiasis. The first functional module prominently clustered eight key protein targets, i.e., TNF, IL6, IL1B, PTGS2, STAT3, NFKB1, HDAC1, and MPO, while PRKCA was included in the third module. These findings indicate that these targets significantly influence the effects of MHR against PF.

### 2.4. Gene Ontology (GO) and Enriched Pathway Analysis

The enrichment analysis identified a total of 87 Kyoto Encyclopedia of Genes and Genomes (KEGG) signaling pathways, along with 295 biological process (BP) terms, 80 molecular function (MF) terms, and 42 cellular component (CC) terms (Appendix A). The bubble plot in Figure 4a displays the top 30 signaling pathways, whereas Figure 4b shows the bubble plot of the top 10 GO terms for BP, MF, and CC. The bubble size and color represent the number of enriched genes and *p*-values, respectively. The darker colors indicate smaller *p*-values, whereas larger bubbles denote higher numbers of enriched therapeutic genes, indicating a stronger association with PF treatment. KEGG pathway enrichment analysis highlighted phenylalanine and arachidonic acid metabolisms as the most significantly enriched pathways, both of which are evidently represented in the KEGG pathway network (Figure 4a).

In the BP category, the top three significant terms were purine-containing salvage, neutrophil apoptotic process, and negative regulation of cardiac muscle relaxation. In the MF category, heparin binding, FMN (Flavin mononucleotide) binding, and chromatin DNA binding were significantly enriched. In the CC category, endolysosome lumen, nuclear envelope lumen, and asymmetric synapse were the most enriched terms.

### 2.5. Investigation of Possible Therapeutic Targets of MHR for PF Treatment

A compound–target network shows interactions between active compounds and potential therapeutic targets (Figure 5). The resulting network consisted of 198 nodes and 698 edges, representing the binding relationships between 86 active compounds and 131 targets. Among the active compounds, those of MHR exhibited the highest degree value (degree = 25), indicating their strong association with multiple core PF targets. Therefore, molecular docking analyses of phytochemicals belonging to MHR with critical therapeutic targets, including TNF, IL6, IL1B, and PTGS2, were performed.

### 2.6. Molecular Docking of Key Targets

The docking results revealed binding affinities for four key targets, viz. TNF, IL6, IL1B, and PTGS2, with binding energies ranging from −4.1 to −9.8 kJ/mol. Augustic acid, nimbolide, and obacunone exhibited the strongest binding affinity with the TNF receptor (−9.1 kJ/mol). While obacunone displayed the best docking score (−7.7 kJ/mol), the IL6 receptor, luteolin, and quercetin showed the highest affinity for the IL1B receptor (−7.5 kJ/mol). On the other hand, apigenin displayed the strongest interaction with the PTGS2 receptor (−9.8 kJ/mol). Table 4 shows the top 21 active compounds with strong binding affinities. Since the 3D structures of 22 compounds are not available in the PubChem database, their detailed information is given in Appendix A. Figure 6 and Figure 7 displayed the molecular docking interactions between active compounds or standard drugs and target proteins. It also illustrates the different types of bonds formed between proteins and their corresponding ligands. In brief, augustic acid bound to the active pockets of TNF and interacted with G121 and T151 to form hydrogen bonds (Figure 6a) while obacunone bound to IL6 via a hydrogen bond interaction with A30, A179, and S37 (Figure 6b). Furthermore, obacunone bound to the active pocket of IL1B and interacted with G64 and S5 to form hydrogen bonds (Figure 7a) while apigenin bound to the active pocket of PTGS2 and interacted with H75, P504, and T341 to form hydrogen bonds (Figure 7b).

### 2.7. Preparation of MHR Extract

In order to maximize the extraction of the phytochemicals from MHR, the mixture of the dried coarse powder of the 21 plants (500 g), belonging to the MHR remedy (according to Table 1), was extracted twice, by maceration, with 70% aqueous ethanol solution instead of pure water, which is traditionally used to prepare the MHR remedy. After solvent evaporation and lyophilization, 125.90 g of the dry powder of MHR extract was obtained (25.18% yield).

### 2.8. Characterization of MHR Extract

The High Performance Liquid Chromatography (HPLC) method was used to determine the fifteen biomarkers of MHR extract (Appendix A). As shown in Figure 8, the chromatogram of MHR extract allowed for the identification of major biomarker peaks. At a wavelength of 254 nm, the retention times for the biomarker compounds are as follows: chebulic acid (**1**) (8.2 min), gallic acid (**2**) (12.8 min), protocatechuic acid (**3**) (22.5 min), bergenin (**4**) (31.8 min), chebulanin (**5**) (46.3 min), corilagin (**6**) (48.1 min), chebulagic acid (**7**) (61.3 min), ellagic acid (**8**) (64.2 min), resveratrol (**9**) (80.4 min), perforatic acid (**10**) (92.9 min), *O*-methyllaloptaeroxyrin (**11**) (104.6 min), rhein (**12**) (105.9 min), loureirin A (**13**) (108.1 min), pectolinarigenin (**14**) (109.7 min), and peucenin-7-methyl ether (**15**) (129.3 min).

MHR extract contained the highest level of chebulagic acid (**7**) at 36.85 mg/g extract, followed by gallic acid (**2**) at 19.79 mg/g extract. Additionally, chebulanin (**5**) (15.94 mg/g extract), corilagin (**6**) (11.27 mg/g extract), ellagic acid (**8**) (10.60 mg/g extract), chebulic acid (**1**) (9.07 mg/g extract), perforatic acid (**10**) (8.86 mg/g extract), loureirin A (**13**) (3.20 mg/g extract), protocatechuic acid (**3**) (2.78 mg/g extract), *O*-methyllaloptaeroxyrin (**11**) (1.47 mg/g extract), and peucenin-7-methyl ether (**15**) (1.26 mg/g extract) were present in smaller amounts. Of note, the content of bergenin (**4**), resveratrol (**9**), rhein (**12**), and pectolinarigenin (**14**) was less than 1 mg/g extract (Table 5).

### 2.9. Effects of MHR Extract and Biomarker Compounds on NO Production in LPS-Induced RAW264.7 Macrophages

MHR extract displayed a dose-dependent reduction of NO production at concentrations ranging from 10 to 100 µg/mL. However, its potency was lower than that of the positive control, indomethacin, at 100 µg/mL. Among the biomarker compounds, resveratrol (**9**) showed the highest inhibition of NO production, with an IC_50_ value of 17.83 μg/mL, followed by rhein (**12**), with an IC_50_ value of 19.68 μg/mL, while ellagic acid (**8**) and loureirin A (**13**) exhibited moderate activity with IC_50_ vales of 42.13 and 72.42 μg/mL, respectively. The rest of the biomarkers, i.e., chebulic acid (**1**), gallic acid (**2**), protocatechuic acid (**3**), bergenin (**4**), chebulanin (**5**), corilagin (**6**), chebulagic acid (**7**), perforatic acid (**10**), *O*-methyllaloptaeroxyrin (**11**), pectolinarigenin (**14**), and peucenin-7-methyl ether (**15**) exhibited no measurable activity (IC_50_ > 100 μg/mL). The positive control, indomethacin, showed an IC_50_ value of 73.42 µg/mL (Figure 9).

Curiously, protocatechuic acid (**3**), chebulanin (**5**), chebulagic acid (**7**), and perforatic acid (**10**) were found to exhibit dose-dependent effects across concentrations of 1–100 µg/mL, whereas pectolinarigenin (**14**) showed dose-dependent effects at concentrations ranging from 10–100 µg/mL. Moreover, chebulic acid (**1**) and gallic acid (**8**) showed dose-dependent effects at concentrations ranging from 50–100 µg/mL.

### 2.10. Cytotoxicity Effects of MHR Extract and Biomarker Compounds on RAW264.7 Macrophages

MHR extract, chebulic acid (**1**), protocatechuic acid (**3**), bergenin (**4**), chebulanin (**5**), corilagin (**6**), chebulagic acid (**7**), perforatic acid (**10**), *O*-methyllaloptaeroxyrin (**11**), and loureirin A (**13**) were all found to be non-cytotoxic within the concentration range of 1–100 µg/mL in the absence of LPS (i.e., the cell viability exceeded 70%). At a concentration of 100 µg/mL, the MHR extract, *O*-methylalloptaeroxylin (**11**), and loureirin A (**13**) did not show a significant difference in cell viability when compared to the positive control, indomethacin. In contrast, gallic acid (**2**), ellagic acid (**8**), resveratrol (**9**), rhein (**12**), pectolinarigenin (**14**), and peucenin-7-methyl ether (**15**) exhibited increased cytotoxicity at higher concentrations, as shown in Figure 10.

## 3. Discussion

The objective of the present study is to integrate network pharmacology with molecular docking approaches to investigate the potential mechanisms of action of MHR remedy for PF treatment. Our findings highlight the multi-target and synergistic effects of the phytochemicals of MHR in modulating inflammatory and immune responses. Literature search revealed that MHR remedy presumably contains 147 bioactive compounds, which were filtered to 86 compounds that possessed high GI absorption and desirable drug-like properties, with each selected compound interacting with multiple therapeutic targets. Our study has identified 8031 genes associated with PF, with 965 overlapping with MHR-related targets. These overlapping genes are involved in critical biological processes, including infection, cell proliferation and differentiation, apoptosis, and inflammation. Notably, four key targets, viz. TNF, IL6, IL1B, and PTGS2 rank among the top ten proteins with the highest degree of connectivity and exhibit strong binding affinities with the 86 compounds of MHR. Subsequently, these key targets were validated by molecular docking analysis.

A combination of network pharmacology and molecular docking has been instrumental in identifying bioactive compounds and targets in various studies [16,17,18]. For example, 726 bioactive compounds were identified from Xiaochaihu Decoction (also known as minor bupleurum decoction), a traditional Chinese medicine (TCM) used for various ailments, by retrieving from the Traditional Chinese Medicine System Pharmacology (TCMSP) database. The decoction was associated with 677 targets overall. Additionally, this study was able to identify 7305 fever-related genes and 400 key targets, which were relevant to fever treatment. Among these, GAPDH (glyceraldehyde 3-phosphate dehydrogenase), AKT1 (serine/threonine kinase 1), INS (insulin), IL6, and VEGFA (vascular endothelial growth factor) were highlighted as major disease-related targets [16]. In another TCM, Bai Hu Tang (White tiger decoction), 120 active compounds were identified, which corresponded to 2176 targets. These findings resulted in the selection of 593 fever-related genes and 34 potential therapeutic targets. Among these, TNF, CASP1 (caspase 1), PTGS2, IL6R (interleukin 6 receptor), CCL4 (C-C motif chemokine ligand 4), HMOX1 (heme oxygenase 1), IL1R2 (interleukin 1 receptor type 2), IFNGR1 (interferon gamma receptor 1), CSF3R (colony stimulating factor 3 receptor), and IFNA21 (interferon alpha 21) were identified as key targets [17]. Another example that used a combination of network pharmacology with a molecular docking approach was Zi Xue Powder. This tool allowed identifying 126 active compounds, corresponding to 341 targets. Additionally, it was possible to retrieve 2951 fever-related genes and to select 707 potential therapeutic targets, among which ALB (albumin), AKT1, INS, TNF, and IL6 were the key targets [18].

Although the previous studies on antipyretic formulas dealt with different active compounds and target genes compared to this study, due to variations in their herbal components, the biological processes involved in their complex mechanisms were similar. Both studies found key roles for processes like cell proliferation, apoptosis, and inflammation [16,17,18]. Although the current study identified a modest number of active compounds, it was possible to identify a substantially larger number of target genes compared to the previous studies. Most importantly, the current study also indicated that relationships of potential therapeutic targets related to phenylalanine and arachidonic acid metabolism pathways, both of which are related to inflammation and fever, were highly enriched.

The close interconnection between the 131 potential therapeutic targets and other key proteins also forms a complex therapeutic network, among which nine core proteins serve as central hubs, playing synergistic roles in the molecular mechanisms underlying infection, inflammation, proliferation, apoptosis, and drug resistance in PF. These genes play crucial roles in biological processes and pathways associated with the anti-inflammatory and therapeutic effects of MHR. These proteins, including enzymes and cytokines, regulate various biological processes such as signal transcription and protein phosphorylation. In particular, TNF, IL6, IL1B, PTGS2, STAT3, and NFKB1 are pivotal in the pathogenesis of fever, further emphasizing their significance as potential therapeutic targets. Although there were considerable efforts to elucidate the mechanisms underlying fever, evidence suggested that its onset is primarily triggered by cytokines such as TNF, IL6, and IL1B. These cytokines activate phospholipase A2 (PLA2), which converts phospholipids into arachidonic acid, and the latter is subsequently metabolized to PGH_2_ (prostaglandin H2) by COX-2 (cyclooxygenase-2), with the increased expression of both COX-2 and mPGES-1 (microsomal prostaglandin E synthase-1) leading to the generation of prostaglandins (PGs) in the central nervous system (CNS). Notably, PGE_2_ (prostaglandin E2) is recognized as a key mediator of fever [50,51].

TNF is also involved in fever induction, as this cytokine can trigger fever depending on IL6 plasma levels but independent of IL1B [52]. However, fever induced by the exogenous TNF is mediated through the release of IL1 in peripheral tissues, but not in the brain [53]. Interestingly, intravenous injection of recombinant human TNF (rhTNF) was able to induce fever in rabbits. These findings link its pyrogenic potential to increased PGE_2_ production, a mechanism associated with glutathione. Recent research further supports this hypothesis, revealing that IL6 binds to IL6 receptors on brain endothelial cells and induces prostaglandin synthase COX-2 expression via the STAT3 signaling pathway. Moreover, IL1 rapidly activates the NF-κB (nuclear factor kappa-light-chain-enhancer of activated B cells) pathway and promotes COX-2 production in cerebral endothelial cells [54], thereby contributing to PGE_2_ synthesis.

GO enrichment analysis was conducted to elucidate the functions of potential therapeutic target proteins and genes, thereby revealing the potential mechanism of MHR in overcoming the resistance to PF treatment. The BPs enriched by MHR in PF were associated with the immune regulation and cardiac function during PF response, all of which are relevant to the identified potential therapeutic targets. Since various factors such as severe infections, chronic inflammation, particularly pneumonia, and mixed infectious etiologies can be common causes of PF, it is necessary to investigate systematically and thoroughly these multiple causative factors [5,6,7,8,9,10]. Therefore, the results obtained in this study suggest that MHR may modulate target proteins involved in the pathophysiology of PF, such as TNF, IL6, IL1B, PTGS2, STAT3, and NFKB1 [55]. The MFs associated with these targets include heparin binding, FMN binding, chromatin DNA binding, and ATPase binding, as well as broader categories like protein, enzyme, and energy substance binding, suggesting their roles in biological metabolism, as supported by the BP and MF results. Furthermore, these functions and BPs are also associated with specific cellular substructures, such as the endolysosome lumen, nuclear envelope lumen, and asymmetric synapse. The CCs associated with inflammatory responses, cell proliferation, RNA transcription, energy conversion, and protein binding further support the therapeutic potential of MHR in PF treatment.

KEGG enrichment analysis revealed multiple signaling pathways implicated in the anti-PF mechanism of MHR. Notably, the core gene targets were enriched in the phenylalanine and arachidonic acid metabolism pathways, indicating the critical importance of these two pathways. These findings suggest that arachidonic acid metabolism may play a crucial role in PF recovery, as depicted in Figure 11. Modulation of these signaling pathways can significantly affect the BPs associated with the core targets. Since these pathways are closely associated with infections, inflammation, proliferation, and apoptosis, they are relevant to PF. Consequently, MHR may exert its anti-PF effects by interfering with these processes, as well as potentially mitigating chemotherapy resistance. The phenylalanine metabolism pathway is a significant pathway in inflammatory diseases, including severe fever [55], and many studies have shown that infections or inflammatory states often lead to substantial increases in the serum phenylalanine [56]. Furthermore, this pathway is associated with various infectious diseases, such as Rocky Mountain spotted fever, viral encephalitis, yellow fever, and pneumococcal and Salmonella infections. The arachidonic acid metabolism pathway is widely understood to be central to fever induction, which is mediated by cytokines like TNF, IL6, and IL1B [50,51]. These cytokines trigger downstream mediators of fever, including the release of arachidonic acid from membrane phospholipids and the activation of COX-2. COX-2 catalyzes the conversion of arachidonic acid to PGH_2_, and subsequent mPGES-1 activity on PGH_2_ results in PGE_2_ production, the ultimate mediator of the febrile response [57,58]. It is plausible that MHR exerts its anti-PF effects, at least in part, through this metabolic pathway.

**Figure 11 pharmaceuticals-18-01541-f011:**
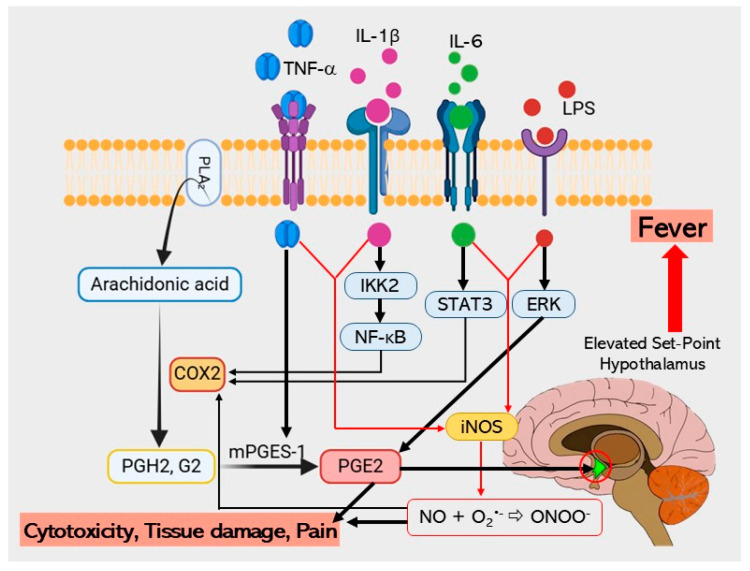
The distribution of genes in the arachidonic acid metabolism pathway. TNF, IL6, and IL1B cytokines trigger downstream mediators of fever, including the release of arachidonic acid from membrane phospholipids and the activation of COX-2 or PTGS2. COX-2 catalyzes the conversion of arachidonic acid to PGH_2_, and subsequent mPGES-1 activity on PGH_2_ results in PGE_2_ production, the ultimate mediator of the febrile response. In response to fever, pro-inflammatory cytokines such as TNF, IL1B, and IL6 induce iNOS expression in macrophages, thereby synthesizing large amounts of NO [59,60,61,62].

According to the ADME (absorption, distribution, metabolism, and excretion) criteria, specifically setting GI absorption = High, and DL = 1,3-methoxyquercetin, apigenin, genkwanin, hispidulin, isorhamnetin, loureirin B, luteolin, *N*-*trans*-feruloyltyramine, *O*-methylalloptaeroxylin, pectolinarigenin, and quercetin possessed the highest number of targets (≥20 targets) within the ADME parameters. Therefore, these compounds could have the highest pharmacological activity in PF. However, it is important to recognize that these compounds are not exclusive to MHR. In fact, the therapeutic effects of medicinal plants are a result of the combined effects of their numerous bioactive compounds rather than the separate action of a few individual components.

All eleven compounds, with the highest number of targets, have already been reported for their relevant pharmacological activities. Thus, 3-methoxyquercetin was shown to inhibit TNF production in RAW 264.7 macrophage cells [59]. The efficacy of apigenin against LPS-induced acute lung injury may be attributed to its primary inhibition of COX-2 and NF-κB gene expression in the lung [60], although it exhibited slight inhibition of TNF-induced JNK (c-Jun-N-terminal kinase) activation [61]. Genkwanin inhibited the activation of the JAK-STAT (janus kinase-signal transducers and activators of transcription) and NF-κB signaling pathways in the synovial tissues of adjuvant-induced arthritis in rats [62]. Hispidulin significantly reduced the levels of NO, reactive oxygen species (ROS), NOS, COX-2, TNF, IL1B, IL6, and PGE_2_ in a dose-dependent manner, suggesting it may inhibit neuroinflammatory responses through NF-κB pathway inhibition [63]. Isorhamnetin contributes to the inhibition of COX-2 expression in response to inflammation [64] and inhibits NF-κB activation [65]. Loureirin B was shown to inhibit the activation of the IL6/STAT3/NF-κB signaling pathway [66]. Luteolin also showed slight inhibition of TNF-induced JNK activation [61]. *N*-*trans*-feruloyltyramine strongly suppressed mRNA expression of iNOS and COX-2, but not TNF, thereby inhibiting NO and PGE_2_ production in LPS-stimulated RAW 264.7 cells [67]. *O*-Methylalloptaeroxylin exhibited the highest inhibition of LPS-induced NO production, with an IC_50_ of 7.92 µg/mL [68]. Pretreatment with pectolinarigenin was shown to inhibit LPS-induced NF-κB activation by interfering with IκB-α (nuclear factor of kappa light polypeptide gene enhancer in B-cells inhibitor alpha) degradation. This compound also mediated NF-κB/Nrf2 (nuclear factor erythroid 2-related factor 2) pathway regulation, which, in turn, inhibited the synthesis of iNOS, COX-2, IL6, IL1B, and TNF in RAW 264.7 and THP1 cells [69]. Quercetin reduced the gene expression of specific factors involved in local vascular inflammation, including IL1R (interleukin-1 receptor), Ccl8 (C-C Motif chemokine ligand 8), IKK (IκB kinase), and STAT3 [70], and inhibited NF-κB activation [71].

Molecular docking revealed the interaction between bioactive compounds and targets in the pharmacological network. Augustic acid exhibited the strongest binding affinity with the TNF receptor. This finding is consistent with the previous research, in which augustic acid showed a marked anti-inflammatory effect, with an ID_50_ of 0.09 mg/ear in a TPA-induced mouse model [72]. Obacunone also exhibited the strongest binding affinity with the IL6 and IL1B receptors. This result was in agreement with the finding by Gao et al. that obacunone displayed anti-inflammatory activity by reducing serum levels of IL6, IL1B, TNF, and IL18 in IL1B-stimulated primary chondrocytes [73]. Additionally, obacunone, isolated from the roots of *Harrisonia perforata*, has been shown to inhibit LPS-induced NO production by RAW 264.7 macrophages, with an IC_50_ of 83.61 µM [32]. It was suggested that apigenin, which showed the strongest binding affinity with the PTGS2 receptor (COX-2), may have a protective effect against LPS-induced acute lung injury, primarily attributed to its inhibition of COX-2 (PTGS2) and NF-κB gene expression in the lung [60].

Biomarker compounds used to characterize MHR extract in this study all possess anti-inflammatory activity. The HPLC chromatogram showed the major peaks of gallic acid (**2**), chebulanin (**5**), corilagin (**6**), chebulagic acid (**7**), ellagic acid (**8**), perforatic acid (**10**), *O*-methyllaloptaeroxyrin (**11**), and peucenin-7-methyl ether (**15**). However, the major compounds in MHR extract were chebulagic acid (**7**), chebulic acid (**1**), gallic acid (**2**), chebulanin (**5**), corilagin (**6**), ellagic acid, and *O*-methyllaloptaeroxyrin (**11**). The isocoumarin, chebulic acid (**1**), coumarins, chebulanin (**5**) and chebulagic acid (**7**), and polyphenols, gallic acid (**2**), corilagin (**6**), and ellagic acid (**8**) are found in the fruit of *P. emblica*, *T. bellirica*, *T. chebula*, and *Terminalia* sp. “Samo Thet in Thai”. Protocatechuic acid (**3**) is a phenolic acid found in *M. ferrea* flower [74]. Bergenin (**4**) is an isocoumarin glycoside found in *F. racemosa* root [75,76]. Resveratrol (**9**) and loureirin A (**13**) are chalcones found in *D. cochinchinensis* wood [77]. Rhein (**12**) is an anthraquinone found in *C. fistula* arils [78]. Perforatic acid (**10**), *O*-methyllaloptaeroxyrin (**11**), and peucenin-7-methyl ether (**15**) are chromones reported from *H. perforata* root [79,80]. Pectolinarigenin (**14**) is an isoflavone from *C. indicum* root [81,82].

Inflammation is one of the mechanisms of fever. In fever reactions, pro-inflammatory cytokines such as TNF, IL1B, IL6, and NF-κB stimulate the expression of iNOS in monocytes or macrophages, neutrophil granulocytes, and many other cells. In the case of bacterial infection, endotoxin is another strong inducer of iNOS expression. Induction of iNOS expression results in the production of large amounts of NO, which can exceed the physiological NO production by up to 1000-fold [83,84,85,86]. The current study revealed that MHR extract, at concentrations ranging from 10–100 µg/mL, exhibited a dose-dependent reduction of NO production in LPS-induced RAW264.7 macrophages. Among the biomarker compounds, resveratrol (**9**), rhein (**12**), ellagic acid (**8**), and loureirin A (**13**) showed good potential to inhibit NO production, while others had no measurable activity (IC_50_ > 100 µg/mL). For their part, protocatechuic acid (**3**), chebulanin (**5**), chebulagic acid (**7**), perforatic acid (**10**), pectolinarigenin (**14**), chebulic acid (**1**), and gallic acid (**2**) all showed a dose-dependent reduction in NO production. Meanwhile, chebulic acid (**1**), chebulanin (**5**), and corilagin (**6**) not only have a low GI absorption but also a zero score of drug-likeness.

Previous studies reported stronger anti-inflammatory activity for some of the compounds used as biomarkers in this study. For example, gallic acid (**2**) was found to reduce the elevated expression of TNF, IL1B, and IL6 induced by LPS [87], while protocatechuic acid has been shown to reduce the production of TNF, IL6, IL1B, and PGE_2_ in LPS-stimulated RAW264.7 cells [88]. Bergenin (**4**) was also found to reduce the production of NO, TNF, IL1B, and IL6 by inhibiting the NF-κB and MAPKs signaling pathways [89], whereas chebulagic (**7**) acid not only inhibited the LPS-induced endothelial cell upregulation of TNF and IL1B in a dose- and time-dependent manner [90] but also inhibited the activation of the NF-κB and MAPK signaling pathways in LPS-stimulated RAW 264.7 cells [91]. *O*-methylalloptaeroxylin (**11**) and pectolinarigenin (**14**) also inhibited NO production in LPS-induced RAW264.7 macrophages, with IC_50_ values of 7.92 and 7.15 µg/mL, respectively [68], while peucenin-7-methyl ether (**15**) exhibited LPS-induced NO production with an IC_50_ of 56.36 µM [33].

Although integrated network pharmacology suggests the TNF signaling pathway and COX pathways may play important roles in fever relief, these findings remain predictive and require further experimental validation. Moreover, while MHR extract demonstrated dose-dependent reduction of NO production in LPS-induced RAW264.7 macrophages, this in vitro model represents only a simplified aspect of the PF microenvironment. Pro-inflammatory cytokines such as TNF, IL6, and IL1B are known to stimulate the gene expression of iNOS, which generates large amounts of NO [89,90]. Consequently, the ability of the MHR extract and its chemical constituents to inhibit NO production serves as a direct indicator of their anti-inflammatory activity. As a life-sustaining process, it is highly complex, and dysregulation has been linked to diverse pathological conditions, including fever. Therefore, additional in vivo and clinical studies will be required to validate these mechanistic insights.

## 4. Materials and Methods

The protocol of this study, with inclusion and exclusion criteria applied in the present study, is summarized by the flowchart in Figure 12.

### 4.1. Screening for Potential Active Compounds and MHR-Related Targets

To identify potential active compounds in MHR, a comprehensive search across electronic databases, including PubMed/Medline, ScienceDirect, ISI Web of Science, ClinicalTrials.gov, and Thai research databases, was conducted, focusing on compounds from the 21 medicinal plants belonging to the MHR remedy with reported anti-inflammatory or antipyretic properties. The identified active compounds were then screened using the SwissADME database (http://www.swissadme.ch/, accessed on 24 November 2023) [92], based on two key ADME indices: GI absorption, which evaluates a compound’s ability to be absorbed in the GI tract after oral administration, and drug-likeness, which assesses the likelihood of a molecule becoming an orally bioavailable drug. The cut-off criteria were set as follows: GI absorption = High, and Drug likeness = 1. Next, the selected compounds were retrieved from the PubChem database (https://pubchem.ncbi.nlm.nih.gov/, accessed on 24 November 2023) [93], to obtain their chemical structures using the Simplified Molecular Input Line Entry System (SMILES). Potential therapeutic targets of MHR were then predicted using the SwissTargetPrediction database (http://www.swisstargetprediction.ch/, accessed on 24 November 2023) [94], with targets having a probability greater than 0 selected for further analysis. A compound–target network was constructed using Cytoscape v3.10.2 to visualize the interactions between active compounds and potential therapeutic targets.

### 4.2. Identification of PF-Related Targets

Three databases were used to identify potential targets related to PF. The GeneCards database (https://www.genecards.org/, accessed on 10 January 2024) [95], provides comprehensive information on all annotated and predicted human genes, integrating data from 150 web sources, including genomic, transcriptomic, proteomic, genetic, clinical, and functional information. The OMIM database (https://omim.org/, accessed on 11 January 2024) [96], serves as a detailed repository of human genes and genetic phenotypes, covering over 16,000 genes and all known Mendelian disorders, with daily updates. Additionally, the PharmGKB database (https://www.pharmgkb.org/, accessed on 12 January 2024) [97], was used to gather pharmacogenomic information on genes associated with PF. Targets identified from these databases were merged and categorized as PF-related targets for further analysis.

### 4.3. Protein-Protein Interaction (PPI) Network and Modular Identification

The Search Tool for the Retrieval of Interacting Genes (STRING) database (https://string-db.org, accessed on 18 February 2024), is widely used in bioinformatics to predict and construct PPI networks [98]. In this study, we employed STRING to analyze the relationships between MHR-related targets and PF-related targets, which were considered as potential therapeutic targets. Significant genes were uploaded to the database, and relevant interaction data were downloaded for further analysis. The PPI network was constructed with “*Homo sapiens*” as the selected species, and a reliability threshold was set to “medium confidence” with a score of ≥0.4, a commonly used parameter for predicting protein interactions. Cytoscape (version 3.10.2), a freely available visualization software, was used to analyze the network topology of relationships among the potential therapeutic targets, including degree value, betweenness centrality, and closeness centrality. Additionally, the Molecular Complex Detection (MCODE) plugin in Cytoscape was utilized to identify central nodes in the network, applying the following parameters: degree cut-off = 2, node score cut-off = 1.0, k-core = 2, and maximum depth = 100.

### 4.4. Functional Enrichment and Pathway Analysis

DAVID Bioinformatics Resources 6.8 (https://davidbioinformatics.nih.gov/, accessed on 15 March 2024) [99], was utilized to analyze all potential therapeutic targets through GO and KEGG pathway enrichment. This analysis aimed to identify related pathways and GO terms, including those in the biological process (BP), molecular function (MF), and cellular component (CC) categories. Pathways and GO terms with *p*-values < 0.05 were considered significant and retained for further analysis. Additionally, the Bioinformatics website (http://www.bioinformatics.com.cn/, accessed on 8 April 2023) was used to visualize the GO and KEGG enrichment analysis results in bar graphs of signaling pathways and bubble plots of GO categories. The network of active compounds, potential targets, and signaling pathways was visualized using Cytoscape v3.10.2 software.

### 4.5. Verification with Molecular Docking

#### 4.5.1. Protein Structures and Modeling of Ligands Preparation

3D crystal structures of four potential human therapeutic targets, viz. TNF (PDB ID: 2AZ5), IL6 (PDB ID: 1ALU), IL1B (PDB ID: 5I1B), and PTGS2 (PDB ID: 3LN1) were obtained from the Brookhaven Protein Data Bank (http://www.rcsb.org/, accessed on 16 May 2024) with a resolution of 2.5 Å. The complete protein structures, along with small-molecule ligand information, were retrieved. For molecular docking, TNF utilized chain protein D [100], while IL6, IL1B, and PTGS2 used chain protein A [100,101,102]. The 3D structures of 86 active compound ligands were sourced from PubChem (https://pubchem.ncbi.nlm.nih.gov/, accessed on 16 May 2024). Prior to docking, water molecules and the original ligands were removed, and the protein structures were saved as PDB files using Discovery Studio 2021 software. AutoDock Tool 1.5.6 was then employed to add hydrogen atoms, calculate charges, and merge nonpolar hydrogens. Both ligands and receptors were stored as PDBQT (Protein Data Bank, Partial Charge, &Atom Type) files, with all flexible bonds in the small molecule ligands set to be rotatable.

#### 4.5.2. Protein-Ligand Docking

Molecular docking was performed using AutoDock Tool 1.5.6 [103], chosen for its high processing speed and superior ability to predict binding models. The conformations of docked and crystal-bound ligands were compared, with docking conducted at 100 num_modes. Grid and box sizes were determined based on literature reviews. The docking site for TNF (PDB: 2AZ5) was defined within a cube of 40 × 40 × 40 Å, covering the ligand-binding site, with a grid point spacing of 1.0 Å, centered at x = −11.9784, y = 70.2727, and z = 14.7429 [100]. For IL6 (PDB: 1ALU), the docking site was set within a cube of 40 × 40 × 40 Å, with a grid point spacing of 1.0 Å, centered at x = −7.677, y = −12.743, and z = 0.048 [102]. The docking site for IL1B (PDB: 5I1B) was defined using a cube of 40 × 40 × 40 Å, with a grid point spacing of 1.0 Å, centered at x = 16.3252, y = 11.4477, and z = 14.0574 [100]. Finally, the docking site for PTGS2 (PDB: 3LN1) was set within a cube of 30 × 30 × 30 Å, with a grid point spacing of 1.0 Å, centered at x = 31.724, y = −22.006, and z = −17.132 [101].

#### 4.5.3. Docking Validation

To validate the docking procedure, the 3D structures of inhibitors for the four human potential therapeutic targets were obtained from PubChem and relevant literature. Docking was conducted using AutoDock Tool 1.5.6, following the same protocols applied to active compounds. The validation ligands included thalidomide (CID: 5426) as an inhibitor of TNF [104], tofacitinib (CID: 9926791) as an inhibitor of IL6 [105], anakinra (CID: 90470007) as an inhibitor of IL1B [106], and celecoxib (CID: 2662) as an inhibitor of PTGS2 [107]. Molecular docking was performed to evaluate the binding interactions between active compounds and target proteins, which were identified based on the intersection of the nine hub proteins and the targets with the highest number of interactions. A binding energy < 0 kJ/mol indicates spontaneous binding between the ligand and receptor, while a binding energy below −4.0 kJ/mol suggests a strong binding affinity.

### 4.6. Preparation of MHR Extract

The roots of *C. micracantha* (CC.MSU-PH-03), *C. indicum* (CC.MSU-PH-04), *F. racemosa* (CC.MSU-PH-05), *H. perforata* (CC.MSU-PH-06), and *T. triandra* (CC.MSU-PH-07) were collected from Roi Et Province, Northeast Thailand, in October 2022. The 11 dried plant materials, including *B. ovata*, *D. cochinchinensis*, *G. chinense*, *L. sinense*, *M. ferrea*, *N. nucifera*, *P. emblica*, *T. hoaensis*, *T. bellirica*, *T. chebula*, and *Terminalia* sp. (Samo Thet in Thai), were purchased from Vejpong Pharmacy (Hock An Tang) Company Limited in Bangkok, Thailand. *A. indica*, *C. fistula*, *T. crispa*, and *V. zizanioides* were purchased from Charoen Sook Osot in Nakhon Pathom Province, Central Thailand. *P. kesiya* was purchased from a Thai traditional pharmacy in Krabi Province, South Thailand. The plant materials were identified by Assoc. Prof. Dr. Somsak Nualkaew, and the voucher specimens were deposited at the Pharmaceutical Chemistry and Natural Products Research Unit, Faculty of Pharmacy, Mahasarakham University, Maha Sarakham, Thailand. Fresh plant materials were cleaned, cut into small pieces, and dried in a hot-air oven at 45 °C for 72 h. After cooling to room temperature, the plant materials were ground into coarse powder. The coarsely powdered plant samples were mixed according to the proportions of the MHR remedy listed in Table 1. The MHR powder (500 g) was macerated in 2500 mL of 70% ethanol (1:5 *w*/*v*) for 3 days, and stirred occasionally. The suspension was then filtered by Whatman No. 1 filter paper, after which the plant residues were macerated in the same condition again and filtered. The filtrates were combined, and the solvent was evaporated by a rotary evaporator at 55–60 °C under a vacuum pressure of ca. 180–200 mbar. The concentrated extract was subsequently lyophilized to obtain a completely dry powder (125.90 g), 25.18% yield.

### 4.7. Chemicals and Reagents

Bergenin, chebulic acid, and chebulagic acid were purchased from Chengdu Alfa Biotechnology (Chengdu, China). Protocatechuic acid, pectolinarigenin, chebulanin, and loureirin A were purchased from Wuhan ChemNorm Biotech (Wuhan, China). Gallic acid, chlorogenic acid, corilagin, ellagic acid, resveratrol, and rhein were purchased from Sigma-Aldrich (St. Louis, MO, USA). Perforatic acid (**10**) (Appendix A), *O*-methyllaloptaeroxyrin (**11**) (Appendix A), and peucenin-7-methyl ether (**15**) (Appendix A) were isolated from *H. perforata* root by Mr. Chinnaphat Chaloemram, Faculty of Pharmacy, Mahasarakham University, Thailand. Commercial-grade 95% ethanol was purchased from ITALMAR company (Bangkok, Thailand). HPLC-grade acetonitrile and AR-grade DMSO were obtained from RCI-Labscan (Bangkok, Thailand). Trifluoroacetic acid (99.9%) was obtained from Acros Organics (Antwerp, Belgium). Indomethacin, lipopolysaccharide from *E. coli* 055:B5 (LPS), and 3-(4,5-dimethylthiazol-2-yl)-2,5-diphenyltetrazolium bro mide (MTT) were purchased from Sigma-Aldrich (MO, USA). Fetal bovine serum (FBS), Penicillin-Streptomycin (P/S), Dulbecco’s Modified Eagle’s Medium (DMEM), Trypan blue stain 0.4%, and Trypsin-EDTA were from Gibco (Waltham, MA, USA).

### 4.8. HPLC Analysis

The HPLC analysis was conducted on an Agilent 1260 Infinity II prime HPLC system (Agilent Technologies, CA, USA). A Luna C18(2) column (5 µm, 100 Å HPLC-grade, 250 × 4.6 mm, Phenomenex^®^ (Phenomenex Inc. VA9525100, Torrance, CA, USA)) was employed for separation, with a flow rate set at 0.8 mL/min, 20 µL, with retention times extending up to 160 min. Chromatographic detection was performed at a wavelength of 254 nm. The mobile phase consisted of 0.1% *v*/*v* trifluoroacetic acid in water (A) and acetonitrile (B) with the following gradient elusion: 2–10% B in 0–30 min, 10–20% B in 30–60 min, 20–30% B in 60–85 min, 30–60% B in 85–120 min, 60–100% B in 120–155 min and holding for 5 min. The peaks were identified by comparing the retention time and UV spectrum with those of the reference standards, viz. chebulic acid (**1**), gallic acid (**2**), protocatechuic acid (**3**), bergenin (**4**), chebulanin (**5**), corilagin (**6**), chebulagic acid (**7**), ellagic acid (**8**), resveratrol (**9**), perforatic acid (**10**), *O*-methyllaloptaeroxyrin (**11**), rhein (**12**), loureirin A (**13**), pctolinarigenin (**14**), and peucen-in-7-methyl ether (**15**) reference standards. The concentrations of marker compounds in the MHR were calculated using the respective standard curves.

### 4.9. Nitric Oxide (NO) Inhibitory Activity and Cytotoxicity of MHR Extract and Biomarkers on RAW 264.7 Macrophages

Murine leukemia macrophage cells (RAW 264.7), obtained from American Type Culture Collection (ATCC TIB-71), were cultured in DMEM complete media containing 10% heat-inactivated FBS and 1% penicillin-streptomycin in a 5% CO_2_ incubator at 37 °C.

The MHR extract and biomarkers were used to investigate the NO inhibitory activity [86]. The cells (1 × 10^5^ cells/well) were seeded in 96-well plates with 100 µL complete DMEM for 24 h, then treated with the sample (1–100 μg/mL) in the presence of 1 µg/mL of LPS, and the plates were subsequently incubated for 24 h. After the incubation, the supernatant 100 µL was assayed for NO using Griess reagent, and the absorbance was measured at 520 nm. The assay was carried out in triplicate, and indomethacin was used as a positive control. The % inhibition was calculated using the equation NO inhibition (%) = [(OD_control_ − OD_sample_)/OD_control_] ×100, where OD_sample_ stands for the optical density of the sample (treated in LPS-induced cells); OD_control_, the optical density of the solvent (treated in LPS-induced cells). The IC_50_ values were calculated using the GraphPad Prism software 10.0.0 (Dotmatics, Boston, MA, USA).

Cytotoxicity of the MHR extract and biomarkers was determined using the MTT assay. Briefly, the plates were incubated at 37 °C in 5% CO_2_ incubator for 24 h. The MTT solution (10 µL, 5 mg/mL in PBS) was added to each well and incubated for 2 h, after which the supernatant was removed and 75 µL of DMSO was added to dissolve the formazan production in cells. The density of formazan solution was measured by a microplate reader at a wavelength of 570 nm. If the cell viability was less than 70%, the sample was considered toxic [32]. Cell viability (%) = [(OD_control_ − OD_sample_)/OD_control_] ×100, where OD_sample_ is the optical density of the non-LPS-induced cells treated with the sample, and OD_control_ is the optical density of the non-LPS-induced cells treated with solvent.

### 4.10. Statistical Analysis

Statistical analysis of network pharmacology was performed by using the bioinformatic tools mentioned above. A *p*-value < 0.05 was considered statistically significant. Chemical contents and pharmacological activity were performed in triplicate. Values for different parameters were expressed as the mean ± standard deviation (SD). The data were statistically analysed using one-way analysis of variance (ANOVA) followed by Tukey’s post-hoc test. The level of significance was set at *p* < 0.01. The prerequisites for conducting this analysis were that the data must be normally distributed, as confirmed by the Shapiro-Wilk analysis, and homogeneous, as demonstrated by Levene’s test. Statistical analysis was performed using SPSS statistical software 16.0.0 (SPSS Inc., Chicago, IL, USA).

## 5. Conclusions

The current study employed an integrated approach of network pharmacology and molecular docking, combined with experimental validation through nitric oxide inhibition assays and HPLC-based chemical profiling, to elucidate the mechanism of action of MHR, a Thai traditional polyherbal remedy. The findings indicate that MHR exerts therapeutic effects on prolonged fever (PF) via a complex network of bioactive compounds, multiple targets, and interconnected pathways. In particular, MHR appears to modulate the arachidonic acid metabolism pathway, involving TNF, IL6, IL1B, and PTGS2, which are central mediators in fever development. These insights not only advance the understanding of PF pathophysiology and the pharmacological basis of MHR but also provide a scientific framework that may streamline future laboratory investigations and clinical studies.

## Figures and Tables

**Figure 1 pharmaceuticals-18-01541-f001:**
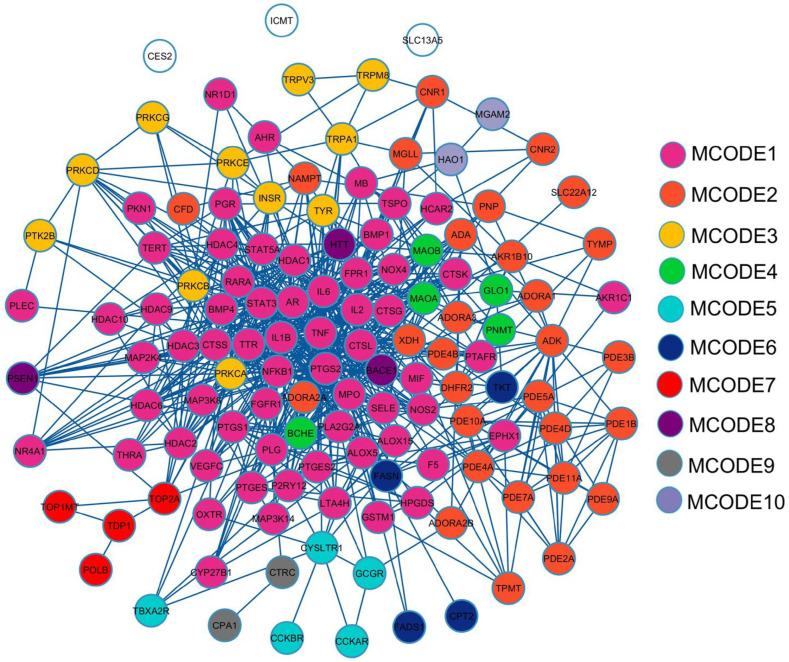
Protein–protein interaction (PPI) network of overlapping genes between MHR and PF-related targets. The circles with different colors represent different MCODE clusters, while uncolored circles indicate nodes not assigned to any MCODE cluster.

**Figure 2 pharmaceuticals-18-01541-f002:**
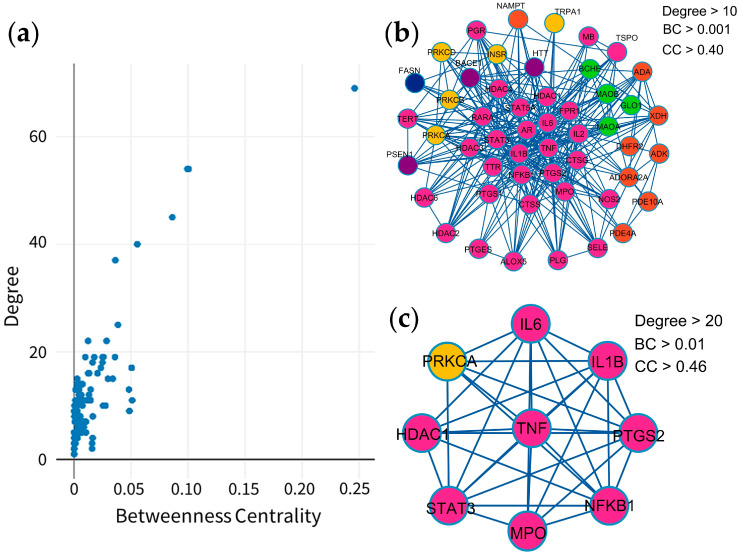
Screening process for hub genes in the PPI network; Scatter plot showing the distribution of degree values (**a**), Identification of genes with a degree > 10 (**b**), Identified 9 hub genes (**c**). Each MCODE network is assigned a unique color.

**Figure 3 pharmaceuticals-18-01541-f003:**
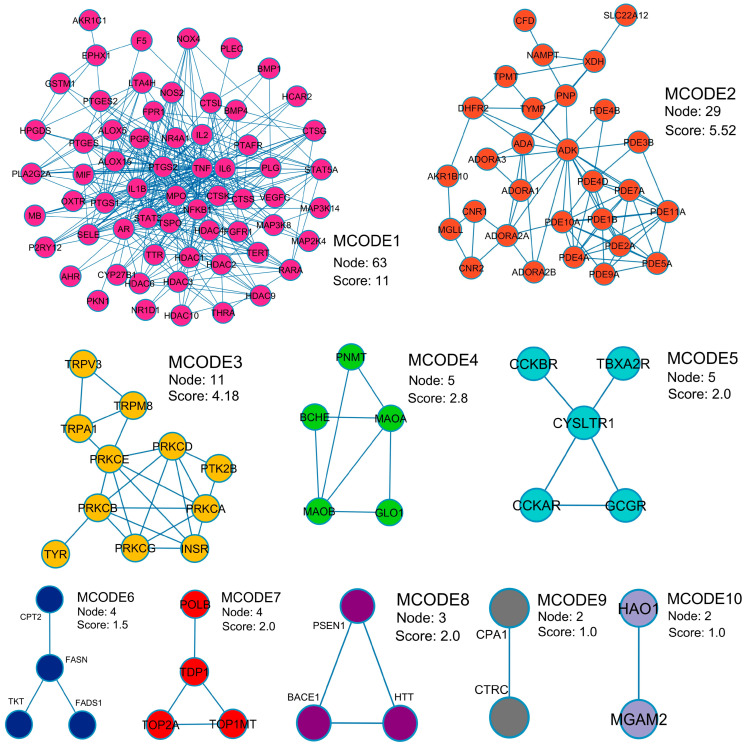
Application of Gene Ontology (GO) enrichment analysis to the MCODE module analysis for sub-network categorization, and each MCODE network is assigned a unique color.

**Figure 4 pharmaceuticals-18-01541-f004:**
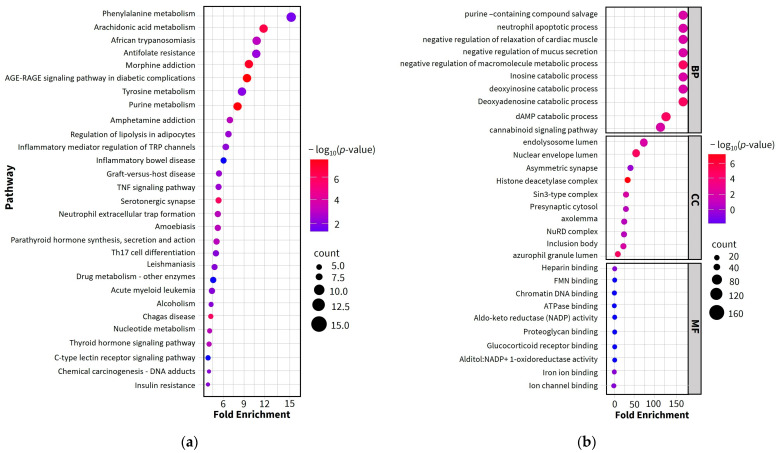
GO function enrichment analysis. The top 30 KEGG pathway enrichment analysis (**a**). The top 10 in terms of biological process (BP), cellular component (CC), and molecular function (MF) (**b**).

**Figure 5 pharmaceuticals-18-01541-f005:**
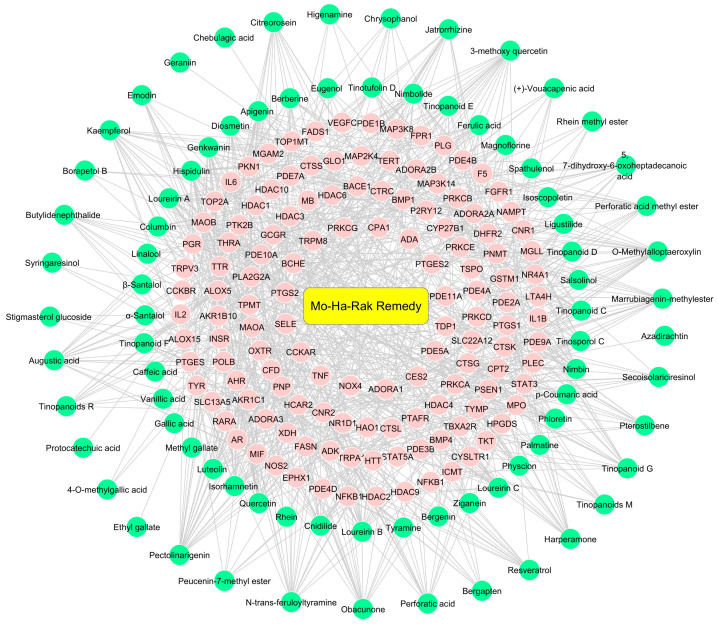
Visualization of the active compounds–potential targets network in PF treatment. Pink rectangles represent genes associated with PF and targets of the active compounds, while green circles denote the plant’s bioactive compounds.

**Figure 6 pharmaceuticals-18-01541-f006:**
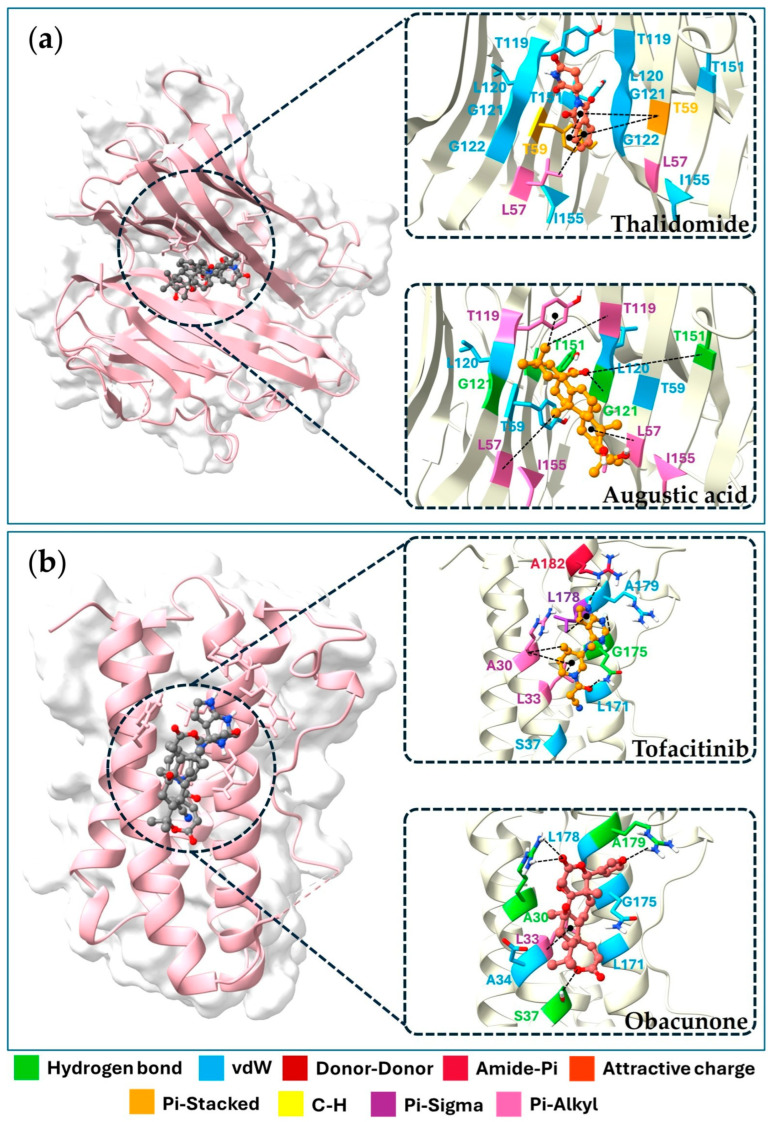
Visualization results of the molecular docking of four target proteins. Molecular docking model of thalidomide and augustic acid with TNF (**a**). Molecular docking model of tofacitinib and obacunone with IL6 (**b**).

**Figure 7 pharmaceuticals-18-01541-f007:**
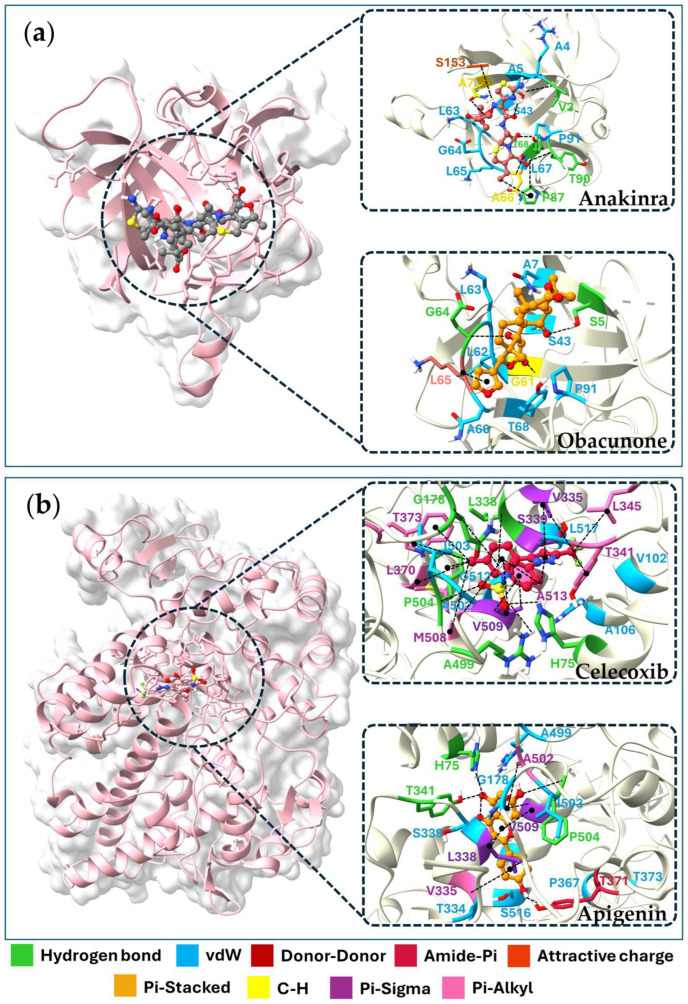
Visualization results of the molecular docking of four target proteins. Molecular docking model of anakinra and obacunone with IL1B (**a**). Molecular docking model of celecoxib and apigenin with PTGS2 (**b**).

**Figure 8 pharmaceuticals-18-01541-f008:**
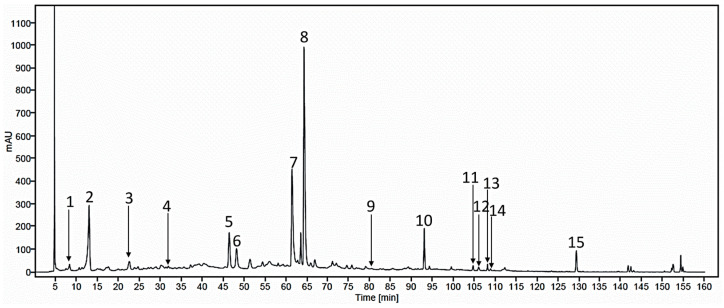
HPLC chromatograms of MHR extract, showing the peaks of biomarker compounds, viz. chebulic acid (**1**), gallic acid (**2**), protocatechuic acid (**3**), bergenin (**4**), chebulanin (**5**), corilagin (**6**), chebulagic acid (**7**), ellagic acid (**8**), resveratrol (**9**), perforatic acid (**10**), *O*-methyllaloptaeroxyrin (**11**), rhein (**12**), loureirin A (**13**), pectolinarigenin (**14**), peucenin-7-methyl ether (**15**).

**Figure 9 pharmaceuticals-18-01541-f009:**
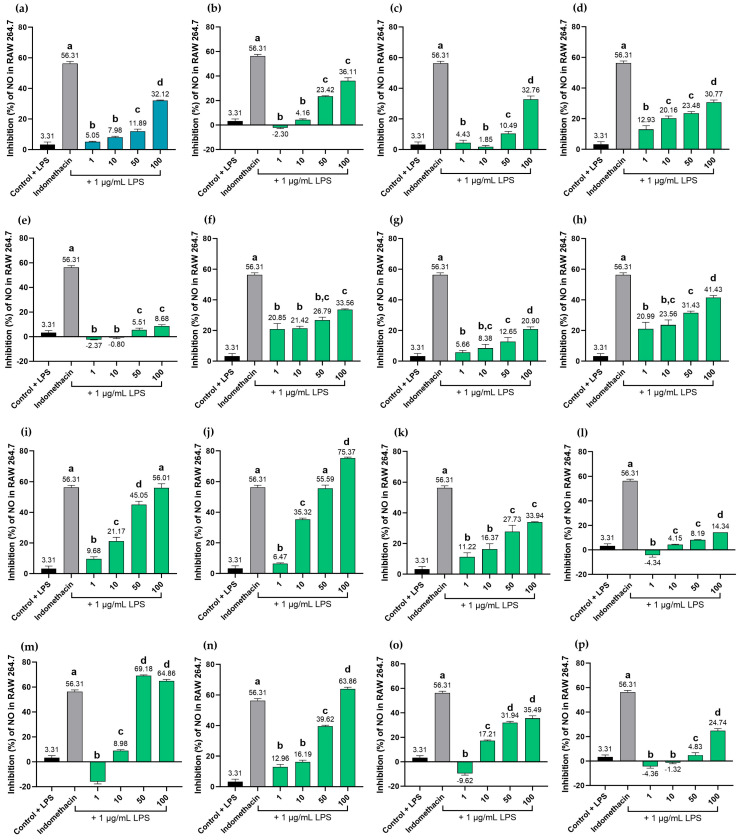
Inhibition of NO production by MHR extract and biomarker compounds at concentrations 1–100 μg/mL on LPS-induced RAW264.7 macrophages; MHR extract (**a**), chebulic acid (**b**), gallic acid (**c**), protocatechuic acid (**d**), bergenin (**e**), chebulanin (**f**), corilagin (**g**), chebulagic acid (**h**), ellagic acid (**i**), resveratrol (**j**), perforatic acid (**k**), *O*-methyllaloptaeroxyrin (**l**), rhein (**m**), loureirin A (**n**), pectolinarigenin (**o**), peucenin-7-methyl ether (**p**). Bars with different letters indicate significant differences (*p* < 0.01). Values are expressed as mean ± SD (n = 3). Indomethacin at 100 μg/mL was used as a positive control.

**Figure 10 pharmaceuticals-18-01541-f010:**
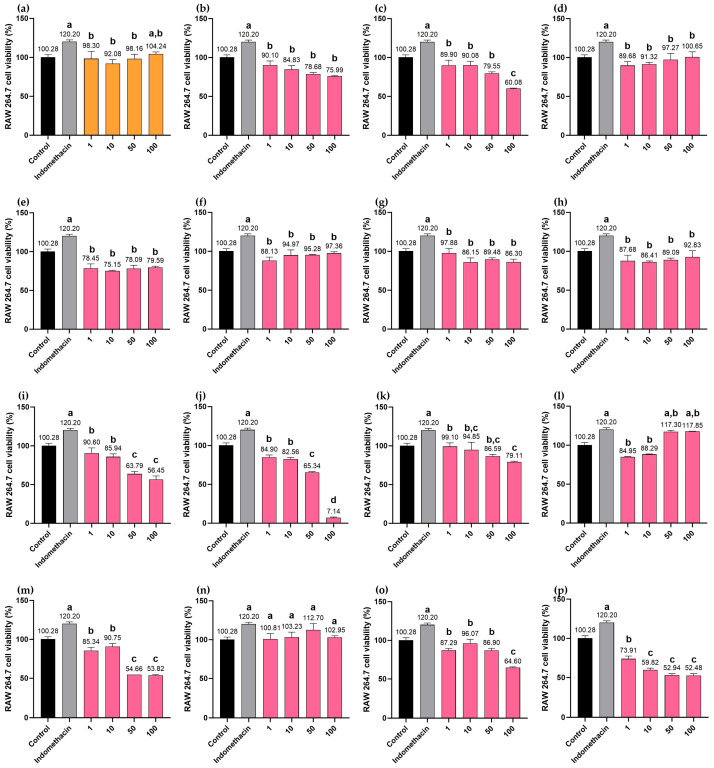
Cell viability of macrophage RAW264.7 treated with MHR extract and biomarker compounds at concentrations 1–100 μg/mL; MHR extract (**a**), chebulic acid (**b**), gallic acid (**c**), protocatechuic acid (**d**), bergenin (**e**), chebulanin (**f**), corilagin (**g**), chebulagic acid (**h**), ellagic acid (**i**), resveratrol (**j**), perforatic acid (**k**), *O*-methyllaloptaeroxyrin (**l**), rhein (**m**), loureirin A (**n**), pectolinarigenin (**o**), peucenin-7-methyl ether (**p**). Bars with the different letters indicate significant differences (*p* < 0.01). Values are expressed as mean ± SD (n = 3). Indomethacin at 100 μg/mL was used as a positive control.

**Figure 12 pharmaceuticals-18-01541-f012:**
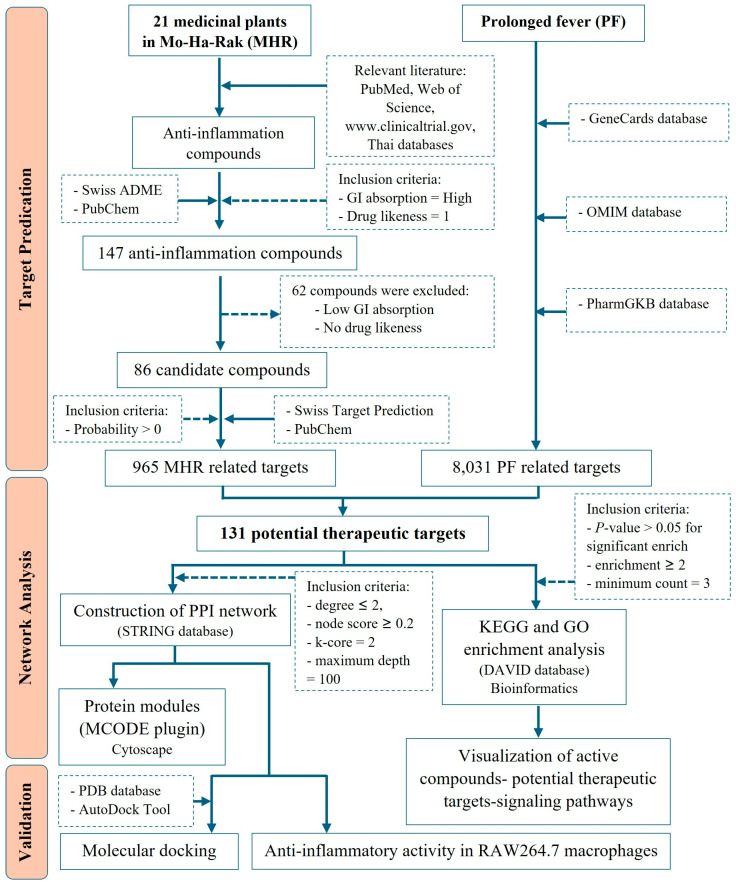
A flowchart of the study protocol to investigate the potential mechanism of MHR in the treatment of PF.

**Table 2 pharmaceuticals-18-01541-t002:** Descriptions of the PPI network hub genes.

Target	Degree	Betweenness Centrality	Closeness Centrality	Type
TNF	69	0.246113	0.658031	Cytokines
IL6	54	0.099402	0.604762	Cytokines
IL1B	54	0.100554	0.604762	Cytokines
PTGS2	45	0.086126	0.561947	Oxidoreductase
STAT3	40	0.055452	0.531381	Transcription
NFKB1	37	0.036233	0.516260	Transcription
HDAC1	25	0.038500	0.484733	Cell cycle
PRKCA	22	0.028542	0.463504	Transferase
MPO	22	0.012275	0.482890	Oxidoreductase

**Table 3 pharmaceuticals-18-01541-t003:** The results of MCODE enrichment analysis.

MCODE	Pathway	Description	Fold Enrichment
MCODE1	hsa00590	Arachidonic acid metabolism	21.3
MCODE1	hsa01523	Antifolate resistance	19.2
MCODE1	hsa05143	African trypanosomiasis	15.6
MCODE2	hsa00230	Purine metabolism	36.3
MCODE2	hsa05032	Morphine addiction	34.0
MCODE2	hsa01232	Nucleotide metabolism	18.2
MCODE3	hsa04960	Aldosterone-regulated sodium reabsorption	85.1
MCODE3	hsa04750	Inflammatory mediator regulation of TRP channels	64.3
MCODE3	hsa05143	African trypanosomiasis	63.8
MCODE4	hsa00360	Phenylalanine metabolism	270.7
MCODE4	hsa00340	Histidine metabolism	196.9
MCODE4	hsa00350	Tyrosine metabolism	180.5
MCODE5	hsa04020	Calcium signaling pathway	27.4
MCODE5	hsa04080	Neuroactive ligand-receptor interaction	23.6
MCODE6	hsa01212	Fatty acid metabolism	114.0
MCODE7	hsa03410	Base excision repair	131.2
MCODE8	hsa05010	Alzheimer disease	15.0

**Table 4 pharmaceuticals-18-01541-t004:** The binding energy of potentially active compounds of MHR and their four target proteins.

Compounds	Binding Energy (ΔG*_bind_*_,_ kcal/mol)			
	TNF (PDB: 2AZ5)	IL6 (PDB: 1ALU)	IL1B (PDB: 5I1B)	PTGS2 (PDB: 3LN1)
**Phytochemicals**				
(+)-Vouacapenic acid	−8.9	−7.3	−6.4	−7.8
Apigenin	−7.5	−6.8	−7.1	−9.8
Augustic acid	−9.1	−7.4	−7.4	−8.5
Berberine	−8.7	−7.4	−6.9	−7.9
Chrysophanol	−8.0	−6.4	−7.4	−9.4
Columbin	−8.7	−7.1	−7.1	−7.4
Diosmetin	−7.6	−6.9	−7.2	−8.9
Emodin	−7.6	−6.7	−7.2	−9.1
Genkwanin	−7.6	−6.5	−7.1	−9.1
Hispidulin	−7.4	−6.9	−7.0	−9.4
Isorhamnetin	−7.6	−6.7	−7.4	−9.7
Jatrorrhizine	−7.7	−6.9	−6.8	−7.0
Kaempferol	−7.3	−6.5	−7.2	−9.5
Luteolin	−7.7	−7.2	−7.5	−9.7
Nimbolide	−9.1	−7.3	−7.0	−8.2
Obacunone	−9.1	−7.7	−8.2	−8.8
Palmatine	−8.0	−6.6	−6.5	−6.7
Perforatic acid methyl ester	−7.4	−6.4	−6.5	−9.3
Quercetin	−7.2	−7.1	−7.5	−9.6
Rhein	−8.3	−6.6	−7.3	−9.2
Stigmasterol glucoside	−8.9	−6.9	−7.4	−8.5
**Standard Drug**				
Thalidomide	−7.4	-	-	-
Tofacitinib	-	−6.4	-	-
Anakinra	-	-	−6.6	-
Celecoxib	-	-	-	−12.1

**Table 5 pharmaceuticals-18-01541-t005:** The contents of the biomarker compounds in the MHR extract.

Marker Compounds	Content of Biomarkers in MHR Extract (mg/g Extract)
Chebulic acid (**1**)	9.07 ± 0.16 ^a^
Gallic acid (**2**)	19.79 ± 0.00 ^b^
Protocatechuic acid (**3**)	2.78 ± 0.02 ^c^
Bergenin (**4**)	0.03 ± 0.00 ^d^
Chebulanin (**5**)	15.94 ± 0.34 ^be^
Corilagin (**6**)	11.27 ± 0.23 ^f^
Chebulagic acid (**7**)	36.85 ± 0.02 ^g^
Ellagic acid (**8**)	10.60 ± 0.01 ^aef^
Resveratrol (**9**)	0.62 ± 0.00 ^h^
Perforatic acid (**10**)	8.86 ± 0.02 ^af^
*O*-Methyllaloptaeroxyrin (**11**)	1.47 ± 0.05 ^i^
Rhein (**12**)	0.31 ± 0.00 ^j^
Loureirin A (**13**)	3.20 ± 0.01 ^k^
Pectolinarigenin (**14**)	0.11 ± 0.00 ^l^
Peucenin-7-methyl ether (**15**)	1.26 ± 0.01 ^i^

Values are expressed as mean ± standard deviation (SD) of triplicate measurements (n = 3). Different superscript letters in the same column indicate significant differences between groups (*p* < 0.01).

## Data Availability

Data is contained within the article and Appendix A.

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
