# Peer review of "Integration of Network Pharmacology, Molecular Docking, and In Vitro Nitric Oxide Inhibition Assay to Explore the Mechanism of Action of Thai Traditional Polyherbal Remedy, Mo-Ha-Rak, in the Treatment of Prolonged Fever"

_pharmaceuticals, 2025, doi:10.3390/ph18101541_

Round 1

Reviewer 1 Report

Comments and Suggestions for Authors

The article “Integration of Network Pharmacology, Molecular Docking and In Vitro Nitric Oxide Inhibition Assay to Explore the Mechanism of Action of Thai Traditional Polyherbal Remedy, Mo-Ha-Rak, in the Treatment of Prolonged Fever” employs modern methods of network pharmacology for elucidation of the mechanism of action of the traditional Thai polyherbal remedy. This topic is crucial for ensuring drug safety by understanding potential herb-drug interactions and increasing the efficacy of herbal medicine by optimising formulations and identifying targeted therapeutic uses. The article is quite interesting, involves appropriate methods, and the results are well presented and described. I would recommend acceptance after minor revision. The points for the revision are presented below.

  1. Introduction. Prolonged fever is also described in children; however, this is not even mentioned in the introduction. Moreover, it is worse to mention whether Mo-Ha-Rak remedy is used for the treatment of children.

  1. Lines 62-64 Mo-Ha-Rak (MHR) is a polyherbal remedy composed of 21 medicinal plants and is used by Thai traditional practitioners for the treatment of PF, poisonous fever (high fever or fever with rash) accompanied by drowsiness, internal body heat, restlessness, delirium, unconsciousness, and internal body inflammation [18].

Here, it should be described how it is usually used in Thai traditional medicine: I presume that the mixture of herbs is extracted somehow; what is the extraction method, mode of drug prescription etc.

  1. Discussion

Line 358: Why are two cytokines underlined? The whole paragraph between lines 358 and 363 is definitely missing references; the role of proinflammatory cytokines is not the finding of this study.

  1. Materials and methods

4.6. Preparation of MHR Extract

In this section, a more detailed description is required. What was done is the preparation of Mo-Ha-Rak, which is a mixture of organs of various plants. Some plants were collected, and some were purchased. It is necessary to indicate the drying method for the collected plants.

The procedure of extraction also requires further clarification:

Lines 604-605 The powder drug (500 g) was macerated in 70% ethanol, 2,500 mL (1:5 w/v) for 3 days in two cycles. What is meant by two cycles? Under which condition was maceration performed?

Lines 606-607 The filtrates were then concentrated using a rotary vacuum evaporator.

Conditions of evaporation, pressure, and temperature should be added.

It is not clear whether evaporation was performed until the solvent completely evaporated. If so, how was the dry residue redissolved, and what was used as a solvent?

Author Response

Reviewer#1: The article “Integration of Network Pharmacology, Molecular Docking and In Vitro Nitric Oxide Inhibition Assay to Explore the Mechanism of Action of Thai Traditional Polyherbal Remedy, Mo-Ha-Rak, in the Treatment of Prolonged Fever” employs modern methods of network pharmacology for elucidation of the mechanism of action of the traditional Thai polyherbal remedy. This topic is crucial for ensuring drug safety by understanding potential herb-drug interactions and increasing the efficacy of herbal medicine by optimising formulations and identifying targeted therapeutic uses. The article is quite interesting, involves appropriate methods, and the results are well presented and described. I would recommend acceptance after minor revision. The points for the revision are presented below.

Reply: We would like to thank reviewer#1’s positive opinion about our mns and giving an insightful appreciation of the subject of our research as well as the correct methodology we have adopted in this research work. Reviewer#1’s comments and suggestions have encouraged us to commit to this complex but vital area of research which still has much more to explore.

Comments 1: Introduction. Prolonged fever is also described in children; however, this is not even mentioned in the introduction. Moreover, it is worse to mention whether Mo-Ha-Rak remedy is used for the treatment of children.

Response 1: We thank reviewer#1 for highlighting this point. Accordingly, we have revised the Introduction to emphasize that prolonged fever (PF) is not restricted to a specific age group, as it can occur in both children and adults. Therefore, the introduction has been revised by adding this information and the dosages used for adults and children in lines 39 and line 67-68 of page 2.

Comments 2: Lines 62-64 Mo-Ha-Rak (MHR) is a polyherbal remedy composed of 21 medicinal plants and is used by Thai traditional practitioners for the treatment of PF, poisonous fever (high fever or fever with rash) accompanied by drowsiness, internal body heat, restlessness, delirium, unconsciousness, and internal body inflammation [18]. Here, it should be described how it is usually used in Thai traditional medicine: I presume that the mixture of herbs is extracted somehow; what is the extraction method, mode of drug prescription etc.

Response 2: We thank reviewer#1 for raising this relevant issue. Accordingly, we have revised the text with a paragraph describing the details of the traditional mode of preparation of MHR remedy, the dosages for adults and children. This added information is in lines 63-68 of page 2.

Comments 3: Discussion

Line 358: Why are two cytokines underlined? The whole paragraph between lines 358 and 363 is definitely missing references; the role of proinflammatory cytokines is not the finding of this study.

Response 3: The text in this paragraph is intended to explain the relationship between the mechanisms of action identified from the integrated network pharmacology study and the inhibition of NO production test of the MHR extract. However, since this issue is also discussed elsewhere, we have now removed this paragraph as suggested by reviewer#1.

Comments 4: Materials and methods 4.6. Preparation of MHR Extract

In this section, a more detailed description is required. What was done is the preparation of Mo-Ha-Rak, which is a mixture of organs of various plants. Some plants were collected, and some were purchased. It is necessary to indicate the drying method for the collected plants. The procedure of extraction also requires further clarification:

Lines 604-605 The powder drug (500 g) was macerated in 70% ethanol, 2,500 mL (1:5 w/v) for 3 days in two cycles. What is meant by two cycles? Under which condition was maceration performed?

Lines 606-607 The filtrates were then concentrated using a rotary vacuum evaporator.

Conditions of evaporation, pressure, and temperature should be added.

It is not clear whether evaporation was performed until the solvent completely evaporated. If so, how was the dry residue redissolved, and what was used as a solvent?

Response 4: We appreciate reviewer#3’s comments to this point. As such, we have revised the method of preparation of MHR extract (4.6 Preparation of MHR Extract), giving all necessary information in detail as suggested by reviewer #1. This information can be found in lines 657 (p.24)-666 (p. 25).

Reviewer 2 Report

Comments and Suggestions for Authors

I read this paper, which explored how the Thai herbal formula MHR works in treating PF. The study used several methods like network pharmacology, molecular docking, and lab experiments. The research was well-designed and the results are meaningful, but there’s still room for improvement. I recommend "major revision". My comments are below:

  1. All the figures are low-resolution. When zoomed in, the text becomes blurry and hard to read. The authors need to improve the resolution and quality of all figures to make sure text and details are clear even when enlarged.
  2. The introduction doesn’t explain network pharmacology well enough. It only says it can be used to study herbal formulas but doesn’t say how. Since network pharmacology has been used for many years and there’s a lot of research on it, I suggest the authors explain its principles, methods, and how it’s used in studying traditional medicine. They should also include recent references (from the last 5 years). For example, these papers used similar methods and could be helpful:

https://doi.org/10.3390/ph18060900

https://doi.org/10.1016/j.phymed.2021.153837

  1. The results section includes a lot of methods details. These should be moved to the "Materials and Methods" section. Please revise thoroughly so that the results part only presents the findings.
  2. The results section also contains discussion content. These parts should be moved to the "Discussion" section. Please make sure results and discussion are clearly separated.
  3. The discussion mentions other similar herbal formulas but doesn’t compare them well with MHR. I suggest the authors analyze the differences and similarities between MHR and other formulas to highlight what makes MHR special and valuable.
  4. The limitations of the study are mentioned but not discussed deeply enough. I recommend adding a more thorough discussion of the study’s limitations.
  5. The conclusion is too long and should be more concise. I suggest focusing on the main findings and what’s new about the research. Briefly summarize how MHR may work against PF and add some future research directions.
  6. Some terms like Prolonged fever (PF) and Mo-Ha-Rak (MHR) are written in both full and short forms multiple times. Please define the abbreviation when it first appears, and use the short form after that. Make sure terminology is used consistently.
  7. The study relies heavily on bioinformatics predictions but doesn’t have enough lab experiments to support them. I suggest giving the authors more time to add experimental validation. If needed, the revision deadline can be extended to make the conclusions stronger and more scientific.
  8. Many references are old, with very few from the last five years. The authors need to update the references and include more recent studies to make the paper more up-to-date and convincing.
Comments on the Quality of English Language

I noticed the English in the paper could be clearer and more professional. I suggested the author use the English editing service from Pharmaceuticals journal to polish the whole text. That way, the paper would meet the journal’s language standards.

Author Response

Reviewer #2:

I read this paper, which explored how the Thai herbal formula MHR works in treating PF. The study used several methods like network pharmacology, molecular docking, and lab experiments. The research was well-designed and the results are meaningful, but there’s still room for improvement. I recommend "major revision". My comments are below. 

Reply: We appreciate Reviewer #2’s positive opinion of our mns especially the experimental design and the results obtained. We are very grateful to Reviewer #2’s comments, which we believe will improve the quality of our paper and its readability.

Comments 1: All the figures are low-resolution. When zoomed in, the text becomes blurry and hard to read. The authors need to improve the resolution and quality of all figures to make sure text and details are clear even when enlarged.

Response 1: All figures in the manuscript have been replaced with high-resolution versions to ensure that the text and details remain clear and legible even when enlarged. 

Comments 2: The introduction doesn’t explain network pharmacology well enough. It only says it can be used to study herbal formulas but doesn’t say how. Since network pharmacology has been used for many years and there’s a lot of research on it, I suggest the authors explain its principles, methods, and how it’s used in studying traditional medicine. They should also include recent references (from the last 5 years). For example, these papers used similar methods and could be helpful: https://doi.org/10.3390/ph18060900

https://doi.org/10.1016/j.phymed.2021.153837

Response 2: We appreciate and agree with reviewer#2’s valuable comments and suggestions. As such, we have revised the “Introduction” accordingly, adding a concise overview of the scope and usefulness of network pharmacology as well as how to perform (target prediction, network construction, topological analysis, pathway enrichment, etc.). We have also provided recent examples from the last five years, including the two papers recommended by reviewer#2 and other papers we have already retrieved during the preparation of the manuscript. The information about network pharmacology has been added in lines 74–85 of page 2.

Comments 3: The results section includes a lot of methods details. These should be moved to the "Materials and Methods" section. Please revise thoroughly so that the results part only presents the findings.

Response 3: We have thoroughly revised the text in all of the subsections of the “Results” section. All detailed procedural descriptions that appeared in the “Results” section of the previous version were moved to the “Materials and Methods” section per your suggestion. The revised sub-sections of the “Results” section are written in red font.

Comments 4: The results section also contains discussion content. These parts should be moved to the "Discussion" section. Please make sure results and discussion are clearly separated.

Response 4: We have proceeded with the section the same manner by removing the excessive discussion therein and add some of them in the “Discussion” section.

Comments 5: The discussion mentions other similar herbal formulas but doesn’t compare them well with MHR. I suggest the authors analyze the differences and similarities between MHR and other formulas to highlight what makes MHR special and valuable.

Response 5: We thank reviewer#2 for his/her interesting perspective. As such, we have revised the Discussion to provide a more detailed comparison between MHR and other similar herbal formulas. “Although previous studies on antipyretic formulas involved different active compounds and target genes due to variations in their herbal components, the underlying biological processes were largely similar. Both the prior studies and the current work highlighted key roles for processes such as cell proliferation, apoptosis, and inflammation [16–18]. Although in our study, a relatively modest number of active compounds was identified, a substantially larger set of target genes was detected compared to previous reports. Notably, our analysis revealed that potential therapeutic targets were highly enriched in phenylalanine and arachidonic acid metabolism pathways, which are directly implicated in inflammation and fever.” This information is added in lines 333–363 (page 17) of the revised manuscript.

Comments 6: The limitations of the study are mentioned but not discussed deeply enough. I recommend adding a more thorough discussion of the study’s limitations.

Response 6: We have now expanded the section on limitations to provide a more thorough consideration. Specifically, we explained that although integrated network pharmacology suggests that the TNF signaling pathway and COX pathways may play important roles in fever relief, these findings remain predictive and require further experimental validation. Moreover, while MHR extract demonstrated dose-dependent reduction of NO production in LPS-induced RAW264.7 macrophages, this in vitro model represents only a simplified aspect of the PF microenvironment. Pro-inflammatory cytokines such as TNF, IL6, and IL1B are known to stimulate iNOS expression, leading to excessive NO generation [89,90]; thus, the ability of MHR to suppress NO production provides supportive but indirect evidence of their anti-inflammatory potential. We also noted that metabolism, as a life-sustaining process, is highly complex, and dysregulation has been linked to diverse pathological conditions, including fever. Therefore, additional in vivo and clinical studies will be required to validate these mechanistic insights. The new discussion appears in lines 531-542 (page 21).

Comments 7: The conclusion is too long and should be more concise. I suggest focusing on the main findings and what’s new about the research. Briefly summarize how MHR may work against PF and add some future research directions.

Response 7: We thank reviewer #2 for his/her suggestion. As such, the conclusion has been shortened and made more concise.

Comments 8: Some terms like Prolonged fever (PF) and Mo-Ha-Rak (MHR) are written in both full and short forms multiple times. Please define the abbreviation when it first appears, and use the short form after that. Make sure terminology is used consistently.

Response 8: We have carefully reviewed the entire manuscript and revised all repetitions of full names and abbreviations. The terminology is now used consistently throughout the manuscript.

Comments 9: The study relies heavily on bioinformatics predictions but doesn’t have enough lab experiments to support them. I suggest giving the authors more time to add experimental validation. If needed, the revision deadline can be extended to make the conclusions stronger and more scientific.

Response 9: We understand reviewer #2’s concern and point of view. In fact, network pharmacology and molecular docking are bioinformatics tools whose integration has become a popular tool now to study traditional medicine formulae and polyherbal remedies. Although it is still more predictive, it has been proven invaluable in probing the targets, genes and enzymes of the biological processes that can interact with bioactive compounds to manifest the pharmacological outcome. As a result, this tool allows us to formulate a hypothesis to design the experimental lab work. Without this tool, we would not have designed and performed the assays we have done. For the experimental work, we are conscious that this work has a very large volume of experimental work. First, the isolation and characterization of the active compounds from the medicinal plants of MHR remedy to serve as biomarker compounds. Secondly, we have performed a HPLC fingerprint analysis of the MHR extract and quantified the bioactive compounds present in the extract. Moreover, we have tested the capacity of MHR extract and all the biomarker compounds for their capacity to inhibit NO production in LPS-Induced RAW264.7 macrophages so that we can link their anti-inflammatory activity using the inhibition of NO production as an endpoint. Finally, MHR and all the biomarker compounds were also assayed for their cytotoxicity against RAW264.7 macrophages. Despite all this huge volume of bioinformatics and experimental work to support the prediction, we are also of the opinion that additional experimental validation would further strengthen the conclusions. Bearing in mind that studying any polyherbal remedies and traditional medicine that involve a holistic approach is not as simple as studying a “one-compound, one-target” approaches. Moreover, the scope of a majority of the papers recently published still relied heavily on bioinformatics predictions. We believe that, with time, this field will progress to more experimental lab work to pinpoint all the hypotheses duly predicted by bioinformatics tools.   

Comments 10: Many references are old, with very few from the last five years. The authors need to update the references and include more recent studies to make the paper more up-to-date and convincing.

Response 10: We have updated the reference list by adding several relevant studies published within the last five years, particularly those related to network pharmacology, molecular docking, nitric oxide inhibition, and the pharmacological activities of polyherbal formulations. These updates ensure that the manuscript reflects the most current research and provides a more convincing scientific context.

 Response to Comments on the Quality of English Language

Point 1: I noticed the English in the paper could be clearer and more professional. I suggested the author use the English editing service from Pharmaceuticals journal to polish the whole text. That way, the paper would meet the journal’s language standards.

Response 1: The English language of the whole manuscript has been revised and refined. This revision has resulted in improved readability and clarity of the revised manuscript.

Round 2

Reviewer 2 Report

Comments and Suggestions for Authors

The author has addressed my comments point-by-point, and I am pleased to recommend the manuscript for acceptance.